# One-Stage Fair Multi-View Spectral Clustering

Rongwen Li
Anhui Provincial International Joint
Research Center for Advanced
Technology in Medical Imaging,
School of Computer Science and
Technology, Anhui University
Hefei, China
e22301284@stu.ahu.edu.cn

Haiyang Hu
Anhui Provincial International Joint
Research Center for Advanced
Technology in Medical Imaging,
School of Computer Science and
Technology, Anhui University
Hefei, China
e23301204@stu.ahu.edu.cn

Liang Du
School of Computer and Information
Technology, Shanxi University
Taiyuan, China
duliang@sxu.edu.cn

Jiarong Chen
Anhui Provincial International Joint
Research Center for Advanced
Technology in Medical Imaging,
School of Computer Science and
Technology, Anhui University
Hefei, China
e23301236@stu.ahu.edu.cn

Bingbing Jiang
School of Information Science and
Technology, Hangzhou Normal
University
Hangzhou, China
jiangbb@hznu.edu.cn

Peng Zhou*
Anhui Provincial International Joint
Research Center for Advanced
Technology in Medical Imaging,
School of Computer Science and
Technology, Anhui University
Hefei, China
zhoupeng@ahu.edu.cn

## Abstract

Multi-view clustering is an important task in multimedia and machine learning. In multi-view clustering, multi-view spectral clustering is one kind of the most popular and effective methods. However, existing multi-view spectral clustering ignores the fairness in the clustering result, which may cause discrimination. To tackle this problem, in this paper, we propose an innovative Fair Multi-view Spectral Clustering (FMSC) method. Firstly, we provide a new perspective of fairness from the graph theory viewpoint, which constructs a relation between fairness and the average degree in graph theory. Secondly, based on this relation, we design a novel fairness-aware regularized term, which has the same form as the ratio cut in spectral clustering. Thirdly, we seamlessly plug this fairness-aware regularized term into the multi-view spectral clustering, leading to our one-stage FMSC, which can directly obtain the final clustering result without any post-processing. We also conduct extensive experiments compared with state-of-the-art fair clustering and multi-view clustering methods, which shows that our method can achieve better fairness.

## CCS Concepts

• **Computing methodologies → Cluster analysis**.

---

*Peng Zhou is the corresponding author.

---

## Keywords

Multi-view clustering, spectral clustering, fair clustering

**ACM Reference Format:**
Rongwen Li, Haiyang Hu, Liang Du, Jiarong Chen, Bingbing Jiang, and Peng Zhou. 2024. One-Stage Fair Multi-View Spectral Clustering. In *Proceedings of the 32nd ACM International Conference on Multimedia (MM '24), October 28-November 1, 2024, Melbourne, VIC, Australia.* ACM, New York, NY, USA, 10 pages. https://doi.org/10.1145/3664647.3681162

## 1 Introduction

Multi-view clustering is a fundamental problem in multimedia processing and machine learning. It aims to learn a consensus clustering result from multiple views or modalities of data and has attracted much attention in recent years [5, 6, 57, 58]. Among these methods, since spectral clustering [9, 44, 47] is a popular and effective graph based method that shows promising performance in clustering on single-view data, it has been widely extended to the multi-view clustering setting. For example, Kumar et al. proposed a multi-view spectral clustering by co-regularizing across multiple views [19]; Li et al. proposed a large-scale multi-view spectral clustering method based on the bipartite graph [21].

Notice that, in real-world scenarios, clustering is often used in applications involving humans such as crime analysis [30] and social networks [40], and thus we should make sure that the clustering result is fair and would not cause any discrimination. In the real world, there are some specific groups, such as females, which may suffer from potential discrimination and need to be protected. These groups are called protected groups. Therefore, one kind of the most important fairness in clustering is group fairness [3], which aims to partition data into several clusters where no clusters contain a disproportionately small number of data in some specific protected groups. Although the existing multi-view spectral clustering methods often achieve good performance on clustering accuracy, none of them consider the fairness of the clustering, and thus may still cause discrimination on some specific groups.

To address this issue, in this paper, we propose a novel fair multi-view spectral clustering method that focuses on the fairness of the result. Since spectral clustering is a graph based method, which tries to find a graph cut to partition data, we also study the fairness from the graph theory perspective. Here we follow a classic definition of group fairness (i.e., Definition 1), which is proposed in [3]. Different from other works that directly use this definition, we observe and construct a new relation between this definition and the *average degree* in graph theory. We theoretically prove that minimizing the average degree can obtain a fair result that follows this definition. Based on this theoretical observation, we design a novel and effective fairness-aware regularized term.

Notice that, since our regularized term is designed from a graph theory viewpoint, it has the same form as spectral clustering. We can naturally and seamlessly integrate this fairness-aware regularized term and the multi-view spectral clustering into a unified framework, leading to a simple yet elegant Fair Multi-view Spectral Clustering (FMSC) method. Notice that conventional multi-view spectral clustering methods are often two-stage methods, which need to learn the continual spectral embedding first and then discretize the spectral embedding to the final clustering result with post-processing like kmeans and spectral rotation. In two-stage methods, the post-processing is separated from the learning method and cannot guarantee the clustering accuracy or fairness. Therefore, the two stages cannot be boosted by each other to achieve a good solution. Different from the two-stage methods, since our fairness-aware regularized term is designed directly from the graph partition perspective, it is a one-stage method that directly obtains the final clustering result from multiple views considering both the clustering accuracy and fairness without any post-processing.

The contributions of this paper are summarized as follows:

- Fair multi-view clustering is an underexplored problem. To the best of our knowledge, we are the first to propose a fair multi-view spectral clustering method.
- We provide a new perspective on fairness. We construct a relation between fairness and average degree to offer a new explanation of fairness from the graph theory viewpoint.
- We carefully design a new fair-awareness regularized term and integrate it into the multi-view spectral clustering seamlessly, forming an elegant and effective fair multi-view spectral clustering framework.
- The extensive experiments on benchmark data sets demonstrate the effectiveness and superiority of the proposed method.

## 2 related works and preliminaries

Throughout this paper, we use a bold uppercase letter and a bold lowercase letter to denote a matrix and a vector, respectively. Given a matrix $\mathbf{M}$, we use $M_{ij}$ to denote its $(i, j)$-th element. We denote $\mathbf{M}_{i\cdot}$ and $\mathbf{M}_{\cdot j}$ as the $i$-th row and column vector of matrix $\mathbf{M}$, respectively.

### 2.1 Multi-View Clustering

Multi-view clustering aims to apply consistency and complementary information of multiple views to learn a consensus clustering result. In recent years, numerous multi-view clustering methods have been proposed [14, 17, 24, 32, 53, 61, 62, 65]. For example, Kang

et al. designed a linear-time large-scale multi-view subspace clustering method based on bipartite graphs [17]; Nie et al. introduced a self-weighted method to determine weights for each view based on their importance [32]; Tao et al. proposed a multi-view clustering approach through ensemble clustering [43]; Liang et al. proposed a robust multi-view clustering approach that can reach the global optimal solution [25]; Zhang et al. proposed a side-constrained multi-view graph clustering method by combining pairwise constraints into a multiple graph fusion framework [55]. Zhou et al. designed a learnable graph filter for multi view clustering [61].

Among these methods, multi-view spectral clustering is one of the most popular methods due to its effectiveness [23, 41, 49, 62, 64]. For example, Tang et al. integrated spectral embedding and k-means into a unified framework to obtain discrete cluster labels [41]; Xia et al. proposed a multi-view spectral clustering method by low-rank and sparse decomposition [49]; Qiang et al. clustered on similarity matrix to obtain discrete indicator matrix [39].

Although these multi-view spectral clustering methods have achieved promising performance, none of them considers the fairness of the clustering result. To tackle this problem, we propose a novel fair multi-view spectral clustering method in this paper.

### 2.2 Spectral Clustering

Let $G(\mathcal{V}, \mathcal{E})$ denote a weighted undirected graph with vertice set $\mathcal{V}$ and edge set $\mathcal{E}$, containing $n$ vertices. Its weights on edges are represented by an adjacency matrix $\mathbf{W} \in \mathbb{R}^{n \times n}$. The objective of spectral clustering is to partition $G$ into $c$ unconnected subgraphs with nonempty vertice subsets $\{\mathcal{X}_1, \mathcal{X}_2, \cdots, \mathcal{X}_c\}$ to minimize the intra-cluster similarity and maximize the inter-cluster similarity. This is achieved by minimizing the Ratio Cut (Rcut) [45] of $G$:

$$Rcut(\mathcal{X}_1, \cdots, \mathcal{X}_c) = \sum_{k=1}^{c} \frac{cut\left(\mathcal{X}_k, \overline{\mathcal{X}}_k\right)}{|\mathcal{X}_k|}, \tag{1}$$

where $\overline{\mathcal{X}}_k$ denotes the complement of set $\mathcal{X}_k$.
$cut\left(\mathcal{X}_k, \overline{\mathcal{X}}_k\right) = \sum_{V_i \in \mathcal{X}_k, V_j \in \overline{\mathcal{X}}_k} w_{ij}$ is a cut of graph $G$, which denotes the summation of the edge weights between $\mathcal{X}_k$ and $\overline{\mathcal{X}}_k$.

Computing $cut\left(\mathcal{X}_k, \overline{\mathcal{X}}_k\right)$ needs a binary indicator vector $\mathbf{y}_k \in \{0, 1\}^n$, where the $i$-th element in $\mathbf{y}_k$ is 1 if the $i$-th vertice $V_i$ belongs to $\mathcal{X}_k$. Denoting a diagonal matrix $\mathbf{D} \in \mathbb{R}^{n \times n}$ as a degree matrix, whose $i$-th diagonal element $d_{ii} = \sum_{j=1}^{n} w_{ij}$, we have

$$cut\left(\mathcal{X}_k, \overline{\mathcal{X}}_k\right) = \sum_{V_i \in \mathcal{X}_k} d_{ii} - cut\left(\mathcal{X}_k, \mathcal{X}_k\right) = \mathbf{y}_k^T \mathbf{D} \mathbf{y}_k - \mathbf{y}_k^T \mathbf{W} \mathbf{y}_k = \mathbf{y}_k^T \mathbf{L} \mathbf{y}_k,$$

where $\mathbf{L} = \mathbf{D} - \mathbf{W}$ is a graph Laplacian matrix [12]. Taking it back to Eq.(1), we can rewrite Rcut as [38] did:

$$Rcut(\mathcal{X}_1, \cdots, \mathcal{X}_c) = \sum_{k=1}^{c} \frac{\mathbf{y}_k^T \mathbf{L} \mathbf{y}_k}{\mathbf{y}_k^T \mathbf{y}_k} = \text{Tr}\left(\mathbf{Y}^T \mathbf{L} \mathbf{Y} \left(\mathbf{Y}^T \mathbf{Y}\right)^{-1}\right)$$

$$= \text{Tr}\left(\left(\mathbf{Y}^T \mathbf{Y}\right)^{-\frac{1}{2}} \mathbf{Y}^T \mathbf{L} \mathbf{Y} \left(\mathbf{Y}^T \mathbf{Y}\right)^{-\frac{1}{2}}\right), \tag{2}$$

where $\mathbf{Y} = [\mathbf{y}_1, \ldots, \mathbf{y}_c] \in \{0, 1\}^{n \times c}$. For the convenience of optimization, let $\mathbf{H} = \mathbf{Y} \left( \mathbf{Y}^T \mathbf{Y} \right)^{-\frac{1}{2}}$ and Eq.(2) can be reformulated as:

$$\min_{\mathbf{H} = \mathbf{Y}(\mathbf{Y}^T\mathbf{Y})^{-\frac{1}{2}}, \mathbf{Y} \in \text{ Ind}} \text{Tr} \left( \mathbf{H}^T \mathbf{L} \mathbf{H} \right), \qquad (3)$$

where Ind is a set of indicator matrices. Eq.(3) is an NP-hard problem [13]. Existing methods [15, 31, 60] relax it by making $\mathbf{H}^T\mathbf{H} = \mathbf{I}$, and disregarding the discrete constraints $\mathbf{Y} \in$ Ind , thereby obtaining the relaxed objective function:

$$\min_{\mathbf{H}^T\mathbf{H}=\mathbf{I}} \text{Tr} \left( \mathbf{H}^T \mathbf{L} \mathbf{H} \right). \qquad (4)$$

The optimal $\mathbf{H}$ consists of the eigenvectors of $\mathbf{L}$ corresponding to the $c$ smallest eigenvalues. After obtaining $\mathbf{H}$, existing methods [16, 37, 50] obtain the final clustering results by some post-processing discretizing methods such as k-means and spectral rotation on $\mathbf{H}$.

## 2.3 Fair Clustering

In recent years, fair clustering has received increasing attention in the artificial intelligence community. Chierichetti et al. first introduced the concept of group fairness in clustering, suggesting that samples within the same protected group should not be clustered together [7]. However, the method of [7] is only applicable to two protected groups. To address this issue, Bera et al. extended the definition of fairness to multivariate protected groups [3], which is shown as follows:

DEFINITION 1. *(Fairness) [3] Let $\mathbf{X} \in \mathbb{R}^{n \times d}$ denote a matrix with $n$ instances and $d$ attributes or features, which are partitioned into $c$ disjoint clusters $C = \{\pi_1, \cdots, \pi_c\}$. Given $t$ disjoint protected groups $\mathcal{G}_1, \mathcal{G}_2, \cdots, \mathcal{G}_t$, let $\eta_i = \frac{|\mathcal{G}_i|}{n}$ and $\eta_i(k) = \frac{|\pi_k \cap \mathcal{G}_i|}{|\pi_k|}$ denote the proportion of group $\mathcal{G}_i$ in the whole data and cluster $\pi_k$, respectively. The fairness of the cluster $\pi_k$ is defined as:*

$$fairness \left( \pi_k \right) = \min \left( \frac{\eta_i}{\eta_i(k)}, \frac{\eta_i(k)}{\eta_i} \right), \ \forall i \in \{1, \cdots t\} \qquad (5)$$

*The fairness of the whole clustering result $C$ is defined as:*

$$fairness(C) = \min_{k \in \{1, \cdots c\}} fairness(\pi_k) \qquad (6)$$

REMARK 1. *$fairness(C) \in [0, 1]$, and the larger $fairness(C)$ is, the fairer the clustering result is. It shows that a fair clustering result requires that the proportion of $\mathcal{G}_i$ in each cluster (i.e. $\eta_i(k)$) should be close to the proportion of $\mathcal{G}_i$ in the whole data (i.e., $\eta_i$). When $\eta_i(k) = \eta_i$, the fairness will achieve its maximum value.*

Based on Definition 1, Kleindessner et al. embedded fairness as a linear constraint into spectral clustering [18]; Chierichetti et al. presented a fair decomposition method which partitioned the data into fair subsets to achieve fair clustering results [7]; Then Backurs et al. accelerated the fairness decomposition to linear complexity [2]; Ghadiri et al. proposed a fair k-means which ensured that all protected groups have the same cluster cost among all clusters [11]; By maximizing and minimizing mutual information, Zeng et al. embedded fairness constraints into deep clustering [52].

Although these works often achieve good performance, they were designed for single-view clustering. Fair multi-view clustering is quite underexplored. Zheng et al. introduced group fairness

into deep multi-view clustering [59]. However, our work is significantly different from [59]. Firstly, [59] applies the view contrastive loss which makes it hard to handle the data with more than two views and ours can naturally handle arbitrarily multiple views. Secondly, [59] defines the fair loss with a soft assignment, which has a gap from the true fairness of Definition 1. Different from this, we propose a new graph theory perspective of fairness and directly optimize the discrete cluster assignment which is equivalent to optimizing the original fairness definition. Our approach is simpler and more effective. Lastly, [59] is a two-stage method, which first learns an embedding and then obtains the final clustering result from the embedding. As analyzed before, the two stages may not be boosted by each other to achieve the optimal goal, and thus our method applies a one-stage method to directly obtain the final clustering result without any post-processing.

## 3 Method

In this section, we introduce our FMSC in more detail.

## 3.1 Fairness-aware Regularized Term

In this paper, we will provide a new perspective of fairness from the graph theory viewpoint. Since fairness is relative to the protected groups according to Definition 1, we first introduce the definition of the *protected group graph* as follows:

DEFINITION 2. *(Protected group graph) Given a data set $\mathcal{X} = \{\mathbf{x}_1, \cdots, \mathbf{x}_n\}$ with $n$ instances and $t$ protected groups $\mathcal{G}_1, \mathcal{G}_2, \cdots, \mathcal{G}_t$ in $\mathcal{X}$, an undirected and unweighted graph $G^p (\mathcal{V}, \mathcal{E}^p)$ with vertice set $\mathcal{V}$ and edge set $\mathcal{E}^p$ is a protected group graph of $\mathcal{X}$ w.r.t. $\mathcal{G}_1, \mathcal{G}_2, \cdots, \mathcal{G}_t$, if $\mathcal{V}$ consists of $n$ vertices corresponding to $n$ instances in $\mathcal{X}$ and there is an edge between the $i$-th and $j$-th vertices if and only if $\mathbf{x}_i$ and $\mathbf{x}_j$ belongs to the same protected group.*

REMARK 2. *Since there is an edge between any instances in the same protected group according to Definition 2, the protected group graph consists of $t$ disconnected complete subgraphs, and each complete subgraph represents a protected group. We call these complete subgraphs as protected group subgraphs. For example, considering 10 people where 5 of them belong to one protected group (e.g. male) and the other 5 people belong to the other protected group (e.g. female), its protected group graph is shown as Figure 1(a), where the circles denote males and the squares denote females.*

According to Definition 1, fair clustering aims to partition $\mathcal{X}$ into $c$ clusters $\pi_1, \cdots, \pi_c$, where the proportion of $\mathcal{G}_i$ in each cluster should be close to each other. From the perspective of the protected group graph, it means that we wish to partition the protected group graph $G^p$ into $c$ cluster subgraphs, where in each cluster subgraph, the proportion of instances in each protected group subgraph should be close to each other. For example, Figure 1(b) shows a fair partition on the protected group graph. The blue vertices form one cluster and the green vertices form the other cluster. In the two clusters, the proportions of instances in two protected group subgraphs are both 1 : 1.

Notice that the protected group subgraph is still a complete subgraph, which means the instances in one protected group subgraph have links to all other instances in the same subgraph but have no links to the ones in other subgraphs. If we wish in each cluster

the proportion of instances in each protected group graph should be close to each other, intuitively, the cluster subgraphs should be "sparse", which means each cluster subgraph should contain as few as possible edges.

In graph theory, *average degree* can be used to evaluate the "sparsity" or "density" of a graph, whose formal definition is as follows:

DEFINITION 3. *(Average degree)* [8] *Let $G(\mathcal{V}, \mathcal{E})$ denote an undirected and unweighted graph. Let $|\mathcal{V}|$ and $|\mathcal{E}|$ be the number of vertices and edges, respectively. The average degree of graph $G$ is defined as:*

$$d_{avg}(G) = \frac{2|\mathcal{E}|}{|\mathcal{V}|} \tag{7}$$

It is easy to verify that the smaller $d_{avg}(G)$ is, the more sparse the graph is. Therefore, intuitively, the smaller the total average degree of all cluster subgraphs is, the fairer the clustering result is. Figure 1 shows a simple toy example. Figure 1(b) and Figure 1(c) show two partitions on the protected group graph in Figure 1(a). In Figure 1(b), according to Definition 1, we have $fairness = 1$, which shows that the results are perfectly fair. Its total average degree of two clusters is $d_{avg}(G) = \frac{4}{4} + \frac{12}{6} = 3$. In Figure 1(c), the $fairness = 0$, which means it is an unfair clustering result. Its total average degree of two clusters is $d_{avg}(G) = \frac{20}{5} + \frac{20}{5} = 8$, which is much larger than the one in Figure 1(b).

More formally, we provide the following Theorem, which constructs a relation between the fairness and the average degree:

THEOREM 1. *Given a protected group graph $G^p(\mathcal{V}, \mathcal{E}^p)$ of data $\mathcal{X}$, we partition it into $c$ cluster subgraphs $G_1^p, \cdots, G_c^p$ by partitioning $\mathcal{X}$ into $c$ clusters $\pi_1, \cdots, \pi_c$. If the partition minimizes the total average degree $\sum_{k=1}^{c} d_{avg}(G_k^p)$, then the corresponding clustering results achieve the highest of the fairness defined in Definition 1.*

PROOF. Let $\mathcal{G}_1, \mathcal{G}_2, \cdots, \mathcal{G}_t$ denote the $t$ protected groups of $G^p$. We now compute $d_{avg}(G_k^p)$ for the $k$-th cluster. Notice that all connective components of $G^p$ are still complete subgraphs, and thus $G_k^p$ also consists of complete subgraphs. Without loss of generality, we assume $G_k^p$ contains $t$ complete subgraphs, and if the number of connected components of $G_k^p$ is $t_1 < t$, we can introduce $t - t_1$ empty subgraphs to make it have $t$ complete subgraphs. In each complete subgraph, there are $|\pi_k \cap \mathcal{G}_i|$ vertices, which is the number of instances in the $k$-th cluster which are in the $i$-th protected group. Since it is a complete subgraph, the number of edges is $\frac{|\pi_k \cap \mathcal{G}_i|(|\pi_k \cap \mathcal{G}_i| - 1)}{2}$. Therefore, the average degree of $G_k^p$ is:

$$d_{avg}(G_k^p) = \frac{\sum_{i=1}^{t} |\pi_k \cap \mathcal{G}_i|(|\pi_k \cap \mathcal{G}_i| - 1)}{|\pi_k|} = \frac{\sum_{i=1}^{t} |\pi_k \cap \mathcal{G}_i|^2}{|\pi_k|} - 1.$$

Then the total average degree is:

$$\sum_{k=1}^{c} d_{avg}(G_k^p) = \sum_{k=1}^{c} \sum_{i=1}^{t} \frac{|\pi_k \cap \mathcal{G}_i|^2}{|\pi_k|} - c = \sum_{i=1}^{t} \sum_{k=1}^{c} \frac{|\pi_k \cap \mathcal{G}_i|^2}{|\pi_k|} - c. \tag{8}$$

Since $c$ is a constant, minimizing $\sum_{i=1}^{c} d_{avg}(G_k^p)$ is equivalent to minimizing $\sum_{i=1}^{t} \sum_{k=1}^{c} \frac{|\pi_k \cap \mathcal{G}_i|^2}{|\pi_k|}$.

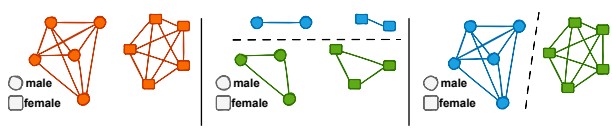

(a) protected group graph    (b) fair partition    (c) unfair partition

**Figure 1: The protected group graph and fair/unfair partition on the protected group graph**

Then, according to Cauchy-Schwarz Inequality, we have

$$\left(\sum_{k=1}^{c} \frac{|\pi_k \cap \mathcal{G}_i|^2}{|\pi_k|}\right)\left(\sum_{k=1}^{c} |\pi_k|\right) \geq \left(\sum_{k=1}^{c} |\pi_k \cap \mathcal{G}_i|\right)^2 = |\mathcal{G}_i|^2, \tag{9}$$

which means

$$\sum_{i=1}^{t} \sum_{k=1}^{c} \frac{|\pi_k \cap \mathcal{G}_i|^2}{|\pi_k|} \geq \sum_{i=1}^{t} \frac{|\mathcal{G}_i|^2}{n}. \tag{10}$$

The equation holds when $\frac{|\pi_1 \cap \mathcal{G}_i|}{|\pi_1|} = \frac{|\pi_2 \cap \mathcal{G}_i|}{|\pi_2|} = \cdots = \frac{|\pi_c \cap \mathcal{G}_i|}{|\pi_c|} = \frac{|\mathcal{G}_i|}{n}$ for any $i$. Notice that, given any cluster $\pi_k$, $\frac{|\pi_k \cap \mathcal{G}_i|}{|\pi_k|} = \eta_i(k)$ and $\frac{|\mathcal{G}_i|}{n} = \eta_i$, where $\eta_i$ and $\eta_i(k)$ are defined in Definition 1. Therefore, we have that minimizing $\sum_{k=1}^{c} d_{avg}(G_k^p)$ will lead to $\eta_i(k) = \eta_i$, and according to Definition 1, this will lead to the maximum of $fairness$. This concludes the proof.                    □

According to Eq.(8), we should minimize $\sum_{i=1}^{t} \sum_{k=1}^{c} \frac{|\pi_k \cap \mathcal{G}_i|^2}{|\pi_k|}$ to obtain the fair clustering result. For simplicity, we denote $\mathbf{Y} \in \{0, 1\}^{n \times c}$ as the cluster indicator matrix, where $Y_{ij} = 1$ if the $i$-th instance belongs to the $j$-th cluster and $Y_{ij} = 0$ otherwise. Similarly, we also define a protected group indicator matrix $\mathbf{P} \in \{0, 1\}^{n \times t}$, where $P_{ij} = 1$ if the $i$-th instance belongs to the $j$-th protected group and $P_{ij} = 0$ otherwise.

Then, it is easy to verify that the $(i, k)$-th element in matrix $\mathbf{P}^T \mathbf{Y}$ is $|\pi_k \cap \mathcal{G}_i|$. Moreover, $\mathbf{Y}^T \mathbf{Y}$ is a diagonal matrix whose $k$-th diagonal element is the number of instances in the $k$-th cluster, which is $|\pi_k|$. To this end, we have

$$\sum_{k=1}^{c} \sum_{i=1}^{t} \frac{|\pi_k \cap \mathcal{G}_i|^2}{|\pi_k|} = \left\| \mathbf{P}^T \mathbf{Y} \left(\mathbf{Y}^T \mathbf{Y}\right)^{-\frac{1}{2}} \right\|_F^2 = \text{Tr}\left(\mathbf{Y}^T \mathbf{P} \mathbf{P}^T \mathbf{Y} \left(\mathbf{Y}^T \mathbf{Y}\right)^{-1}\right) \tag{11}$$

Therefore, we obtain our fair-awareness regularized term $\text{Tr}\left(\mathbf{Y}^T \mathbf{P} \mathbf{P}^T \mathbf{Y} \left(\mathbf{Y}^T \mathbf{Y}\right)^{-1}\right)$. According to Theorem 1, minimizing this regularized term, we can achieve a fair clustering result.

### 3.2 Fair Multi-View Spectral Clustering

Since our designed fairness-aware regularized term Eq.(11) has the same form as Rcut defined in Eq.(2), we can naturally and seamlessly integrate this regularized term into spectral clustering. Notice that conventional spectral clustering methods often optimize Eq.(4) to learn a continuous embedding $\mathbf{H}$ and then discretize $\mathbf{H}$ to obtain the final cluster indicator matrix $\mathbf{Y}$. As analyzed before, this two-stage method separates the embedding learning and discretizing, and thus the two stages cannot be boosted by each other to achieve a good solution.

Different from the two-stage method, since our fairness-aware regularized term involves the discrete cluster indicator matrix $\mathbf{Y}$,

we try to directly solve the original Rcut instead of the continuous approximation.

In more detail, we denote a multi-view data set as $\mathcal{X} = \{\mathbf{X}^{(1)}, \mathbf{X}^{(2)}, \cdots, \mathbf{X}^{(v)}\}$ with $v$ views, where $\mathbf{X}^{(i)} \in \mathbb{R}^{n \times d^{(i)}}$ is the feature matrix of the $i$-th view containing $n$ instances and $d^{(i)}$ features. For the $i$-th view, we need to construct its Laplacian matrix $\mathbf{L}^{(i)}$. We first compute the similarity matrix $\mathbf{S}^{(i)} \in \mathbb{R}^{n \times n}$ of the $i$-th view, whose $(p, q)$-th element is $S_{pq}^{(i)} = e^{-\frac{\left\|\mathbf{x}_{p.}^{(i)} - \mathbf{x}_{q.}^{(i)}\right\|_2^2}{2\sigma^2}}$, where $\sigma$ is a bandwidth parameter and is set as the median of the Euclidean distances of all instance pairs. Then we construct the $k$-NN graph from $\mathbf{S}^{(i)}$ whose adjacency matrix $\mathbf{W}^{(i)}$. If $\mathbf{x}_p$ is a neighbor of $\mathbf{x}_q$ or $\mathbf{x}_q$ is a neighbor of $\mathbf{x}_p$ in the $i$-th view, then $W_{pq}^{(i)} = S_{pq}^{(i)}$, and $W_{pq}^{(i)} = 0$ otherwise. For simplicity, we fix the number of neighbors $k = 5$ in the $k$-NN graph.

Given $\mathbf{W}^{(i)}$, we can compute its Laplacian matrix. Here, we use the normalized Laplacian: $\mathbf{L}^{(i)} = \mathbf{I} - \mathbf{D}^{-\frac{1}{2}} \mathbf{W}^{(i)} \mathbf{D}^{-\frac{1}{2}}$, where $\mathbf{D}$ is a diagonal matrix whose diagonal element $D_{pp} = \sum_{q=1}^{n} W_{pq}^{(i)}$.

Then we combine the Laplacian matrices of all views with weight $\alpha_i^2 \in [0, 1]$ to obtain the consensus Laplacian matrix $\sum_{i=1}^{v} \alpha_i^2 \mathbf{L}^{(i)}$. Take this consensus Laplacian matrix into Eq.(2), we obtain the discrete multi-view spectral clustering:

$$\min_{\mathbf{Y}, \boldsymbol{\alpha}} \quad \mathrm{Tr}\left(\mathbf{Y}^T \left(\sum_{i=1}^{v} \alpha_i^2 \mathbf{L}^{(i)}\right) \mathbf{Y} \left(\mathbf{Y}^T \mathbf{Y}\right)^{-1}\right),$$
$$s.t. \quad \mathbf{Y} \in \{0, 1\}^{n \times c}, \sum_{k=1}^{c} Y_{pk} = 1, \sum_{i=1}^{v} \alpha_i = 1, \alpha_i \geq 0. \quad (12)$$

Taking our fairness-aware regularized term Eq.(11) into the discrete multi-view spectral clustering, we obtain the final objective function of our FMSC:

$$\min_{\mathbf{Y}, \boldsymbol{\alpha}} \quad \mathrm{Tr}\left(\mathbf{Y}^T \left(\sum_{i=1}^{v} \alpha_i^2 \mathbf{L}^{(i)} + \lambda \mathbf{P} \mathbf{P}^T\right) \mathbf{Y} \left(\mathbf{Y}^T \mathbf{Y}\right)^{-1}\right),$$
$$s.t. \quad \mathbf{Y} \in \{0, 1\}^{n \times c}, \sum_{k=1}^{c} Y_{pk} = 1, \sum_{i=1}^{v} \alpha_i = 1, \alpha_i \geq 0, \quad (13)$$

where $\lambda$ is a balance hyper-parameter to control the trade-off between the clustering accuracy and the fairness. Larger $\lambda$ will lead to a fairer clustering result.

## 3.3 Optimization

The problem in Eq.(13) involves two variables, $\mathbf{Y}$ and $\boldsymbol{\alpha}$. We provide an alternative algorithm for optimization.

*3.3.1 Optimization of $\mathbf{Y}$.* When fixing $\boldsymbol{\alpha}$, the subproblem w.r.t. $\mathbf{Y}$ can be rewritten as follows:

$$\min_{\mathbf{Y}} \sum_{k=1}^{c} \frac{\mathbf{Y}_{.k}^T \mathbf{B} \mathbf{Y}_{.k}}{\mathbf{Y}_{.k}^T \mathbf{Y}_{.k}} \quad s.t. \ \mathbf{Y} \in \{0, 1\}^{n \times c}, \sum_{k=1}^{c} Y_{pk} = 1 \quad (14)$$

where $\mathbf{B} = \sum_{i=1}^{v} \alpha_i^2 \mathbf{L}^{(i)} + \lambda \mathbf{P} \mathbf{P}^T$. Notice that there is only one 1 in each row of $\mathbf{Y}$, and thus we can solve $\mathbf{Y}$ row by row. When optimizing the $i$-th row, we replace this row with $[1, 0, \ldots, 0]$, $[0, 1, 0, \ldots, 0]$, ..., $[0, \ldots, 0, 1]$, respectively, and find the one which leads to the minimum of the objective function. Then we set the $i$-th row as this row vector. Wang et al. provide an efficient way to optimize the $i$-th row of $\mathbf{Y}$ by reducing the computation redundancy [46].

In more detail, when optimizing the $i$-th row, we first compute the current $\{\mathbf{Y}_{.k}^T \mathbf{B} \mathbf{Y}_{.k}\}_{k=1}^{c}$ and $\{\mathbf{Y}_{.k}^T \mathbf{Y}_{.k}\}_{k=1}^{c}$, and save these values for the later calculation. As suggested by [46], we introduce some auxiliary matrices with the same size as $\mathbf{Y}$: $\{\mathbf{Y}^{(s)}\}_{s=1}^{c}$ and $\mathbf{Y}^{(0)}$, where $\mathbf{Y}^{(s)}$ is a binary matrix with only the $s$-th element in the $i$-th row is 1 and the rest elements of the $i$-th row are 0s, and all elements in the $i$-th row of $\mathbf{Y}^{(0)}$ are 0s. The other rows (i.e., except the $i$-th row) of $\mathbf{Y}^{(s)}$ and $\mathbf{Y}^{(0)}$ are identical to $\mathbf{Y}$. We can see that, minimizing Eq.(14) is to find a $\mathbf{Y}^{(s)}$ to minimize the objective function. Notice that only the $i$-th rows in $\{\mathbf{Y}^{(s)}\}_{s=1}^{c}$ are different from each other and other rows are the same. Therefore, to find the optima $\mathbf{Y}^{(s)}$, we only need to compute the different parts in the objective function [46], which is:

$$\mathcal{L}\left(\mathbf{Y}^{(s)}\right) = \frac{\mathbf{Y}_{.s}^{(s)T} \mathbf{B} \mathbf{Y}_{.s}^{(s)}}{\mathbf{Y}_{.s}^{(s)T} \mathbf{Y}_{.s}^{(s)}} - \frac{\mathbf{Y}_{.s}^{(0)T} \mathbf{B} \mathbf{Y}_{.s}^{(0)}}{\mathbf{Y}_{.s}^{(0)T} \mathbf{Y}_{.s}^{(0)}}, \quad (15)$$

Assuming that the $m$-th element in the $i$-th row of the current $\mathbf{Y}$ is 1, to compute $\mathcal{L}\left(\mathbf{Y}^{(s)}\right)$, we need to discuss two cases whether $s$ is equal to $m$. If $s = m$, we have $\mathbf{Y}^{(s)} = \mathbf{Y}$ and $\mathbf{Y}_{.s}^{(0)} = \mathbf{Y}_{.s} - \boldsymbol{\delta}$, where $\boldsymbol{\delta} \in \mathbb{R}^n$ is a column vector with the $i$-th element being 1 and the rest elements being 0s. Therefore, Eq.(15) can be computed as:

$$\mathcal{L}\left(\mathbf{Y}^{(s)}\right) = \frac{\mathbf{Y}_{.s}^T \mathbf{B} \mathbf{Y}_{.s}}{\mathbf{Y}_{.s}^T \mathbf{Y}_{.s}} - \frac{\mathbf{Y}_{.s}^T \mathbf{B} \mathbf{Y}_{.s} - 2\mathbf{Y}_{.s}^T \mathbf{B}_{.i} + B_{ii}}{\mathbf{Y}_{.s}^T \mathbf{Y}_{.s} + 1} \quad (16)$$

If $s \neq m$, we have $\mathbf{Y}_{.s}^{(0)} = \mathbf{Y}_{.s}$ and $\mathbf{Y}_{.s}^{(s)} = \mathbf{Y}_{.s} + \boldsymbol{\delta}$. Eq.(15) can be calculated as:

$$\mathcal{L}\left(\mathbf{Y}^{(s)}\right) = \frac{\mathbf{Y}_{.s}^T \mathbf{B} \mathbf{Y}_{.s} + 2\mathbf{Y}_{.s}^T \mathbf{B}_{.i} + B_{ii}}{\mathbf{Y}_{.s}^T \mathbf{Y}_{.s} + 1} - \frac{\mathbf{Y}_{.s}^T \mathbf{B} \mathbf{Y}_{.s}}{\mathbf{Y}_{.s}^T \mathbf{Y}_{.s}}. \quad (17)$$

Notice that, $\{\mathbf{Y}_{.s}^T \mathbf{B} \mathbf{Y}_{.s}\}_{s=1}^{c}$ and $\{\mathbf{Y}_{.s}^T \mathbf{Y}_{.s}\}_{k=1}^{c}$ have already been computed at first and do not need to be computed anymore. Therefore, Eq. (16) or Eq.(17) can be computed efficiently. Then, we obtain the optimal value of $\mathbf{Y}^{(s)}$, denotes as $\mathbf{Y}^{(s^*)}$, where $s^*$ represents the optimal position of element 1 in the $i$-th row. We set $\mathbf{Y} = \mathbf{Y}^{(s^*)}$.

If $s^* = m$, it means that $\mathbf{Y}$ does not change, and we can directly compute the $(i+1)$-th row. If $s^* \neq m$, we need to update the current $\mathbf{Y}_{.k}^T \mathbf{B} \mathbf{Y}_{.k}$ and $\mathbf{y}_k^T \mathbf{y}_k$ for $k \in \{m, s^*\}$ before optimizing the $(i+1)$-th row. We can update these values as follows:

$$\mathbf{Y}_{.m}^T \mathbf{B} \mathbf{Y}_{.m} = \mathbf{Y}_{.m}^{(0)T} \mathbf{B} \mathbf{Y}_{.m}^{(0)}, \ \mathbf{Y}_{.m}^T \mathbf{Y}_{.m} = \mathbf{Y}_{.m}^{(0)T} \mathbf{Y}_{.m}^{(0)}, \quad (18)$$

$$\mathbf{Y}_{.s^*}^T \mathbf{B} \mathbf{Y}_{.s^*} = \mathbf{Y}_{.s^*}^{(s^*)T} \mathbf{B} \mathbf{Y}_{.s^*}^{(s^*)}, \ \mathbf{Y}_{.s^*}^T \mathbf{Y}_{.s^*} = \mathbf{Y}_{.s^*}^{(s^*)T} \mathbf{Y}_{.s^*}^{(s^*)}, \quad (19)$$

Notice that these updations are also efficient because all values on the right-hand sides have already been computed.

*3.3.2 Optimization of $\boldsymbol{\alpha}$.* When fixing $\mathbf{Y}$, we have following subproblem w.r.t. $\boldsymbol{\alpha}$:

$$\min_{\boldsymbol{\alpha}} \sum_{i=1}^{v} \alpha_i^2 e_i, \quad s.t. \sum_{i=1}^{v} \alpha_i = 1, \ \alpha_i \geq 0, \quad (20)$$

where $e_i = \mathrm{Tr}\left(\mathbf{Y}^T \mathbf{L}^{(i)} \mathbf{Y} \left(\mathbf{Y}^T \mathbf{Y}\right)^{-1}\right)$, according to Cauchy-Schwarz Inequality, we obtain the closed-form solution of $\alpha_i$ as:

$$\alpha_i = \frac{e_i^{-1}}{\sum_{j=1}^{v} e_j^{-1}}. \quad (21)$$

---

**Algorithm 1** Fair Multi-View Spectral Clustering

---

**Input:** Multi-view data $\mathcal{X} = \{\mathbf{X}^{(1)}, \mathbf{X}^{(2)}, \cdots, \mathbf{X}^{(v)}\}$, protected groups $\mathcal{G}_1, \cdots, \mathcal{G}_T$, fairness hyper-parameter $\lambda$.

1: Construct the Laplacian matrix $\mathbf{L}^{(i)}$ for each view. Construct the one-hot protected group matrix $\mathbf{P}$.
2: Initialize $\alpha_i = \frac{1}{v}$.
3: **repeat**
4:     Update $\mathbf{Y}$ by solving Eq.(14).
5:     Update $\boldsymbol{\alpha}$ by Eq.(21).
6: **until** Converges

**Output:** The final partition matrix $\mathbf{Y}$.

---

## 3.4 Algorithm and Discussion

Algorithm 1 summarizes the process of our FMSC. The detailed pseudo-code of the algorithm is shown in Appendix. Both updating $\mathbf{Y}$ or $\boldsymbol{\alpha}$ make the objective function decrease monotonically and it has a lower bound. Therefore, the algorithm can always converge. In practice, it often converges very fast (often within ten iterations).

Now we analyze the time complexity. Since we need to construct $k$-NN graphs, the time complexity is $O\left(n^2 dv\right)$. Optimizing $\mathbf{Y}$ has a time complexity of $O\left(n^2 c\right)$ [46]. Computing $\boldsymbol{\alpha}$ has a complexity of $O(n)$. Therefore, the bottleneck is to construct the $k$-NN graph whose time complexity is $O\left(n^2 dv\right)$. Although the time complexity seems a little high, it is often fast in practice and there exist many speedup methods for the $k$-NN graph construction.

## 4 Experiments

In this section, we conduct experiments to demonstrate the effectiveness of our proposed method.

## 4.1 Data Sets

We conduct experiments on eight real and synthetic fair multi-view data sets, including Yale [4], ORL [27], COIL [1], Scene [2], Jaffe [29], Har [1], HCV [3], and Credit Card [59]. Yale and ORL are two multi-view face image data sets for fair clustering. In Yale, individuals wearing glasses form one protected group, while those without glasses form the other protected group as [20] did. In ORL, variations in a person's facial orientation, e.g., facing front or slightly sideways, form different protected groups [20]. Jaffe, Har, HCV, and Credit Card are single-view data sets for fair clustering. We generate multiple views as the previous works did. Specifically, for Credit Card and HCV, following [59], we generate two views using non-linear functions (i.e., Sigmoid and Relu). For Jaffe, besides the original view, we employ pre-trained ResNet-50, VGG-16, and autoencoders with various hidden layer sizes to obtain five additional views [35]. Similar to Jaffe, we generate two views for Har by using two autoencoders with different sizes of the hidden layer and form the three views together with the original view as [10] did. For natural multi-view data sets COIL and Scene, following [51], we randomly assign each instance to a protected group with a Bernoulli distribution whose $p = 0.5$ to form two protected groups. Details of the data sets are shown in Table 1.

---

[1]https://www.cs.columbia.edu/CAVE/software/softlib/coil-20.php
[2]https://figshare.com/articles/dataset/15-Scene_Image_Dataset/7007177
[3]https://archive.ics.uci.edu/dataset/571/hcv+data

---

**Table 1: Description of the data sets.**

| Data sets | #Samples | #Clusters | #Features | Protected Groups |
|---|---|---|---|---|
| Yale | 165 | 15 | 9/512/50 | w/o glasses (2) |
| Har | 10299 | 6 | 400/1000/561 | Person Identity (30) |
| ORL | 400 | 40 | 4096/3304/6750 | Facial Orientation (2) |
| Credit Card | 5000 | 5 | 22/22 | Gender (2) |
| Jaffe | 213 | 10 | 200/400/600/800/1000/676 | Expression (7) |
| Hcv | 615 | 5 | 13/13 | Gender (2) |
| COIL | 1440 | 20 | 1024/3304/6750 | Synthetic Binary (2) |
| Scene | 4485 | 15 | 20/59/40 | Synthetic Binary (2) |

## 4.2 Experimental Setup

We compare our method with 9 state-of-the-art multi-view clustering methods including AMGL [33], AWP [34], CGD [42], COMVSC [56], SMSC [15], OPLFMVC [28], CGL [22], EMVGC [48], and RCAGL [26]; a deep fair multi-view clustering method Fair-MVC [59]; and 3 single-view fair clustering methods including SpFC [18], VFC [63], and FFC [36]. For the single-view fair clustering methods, we concatenate the features of all views to form one single view and run them on this view. In our method, we tune $\lambda$ in the range $[10^{-5}, 10^1]$. For other methods, we tune the hyper-parameters as suggested in their respective papers. For all methods on all data sets, the number of clusters is set as the true number of classes. We utilize Accuracy (ACC) and Normalized Mutual Information (NMI) to evaluate the clustering performance. Besides, we also employ balance (Bal) [54] and Minimal Normalized Conditional Entropy (MNCE) [52] to evaluate fairness. Specifically, Bal is defined as:

$$\text{Bal}(C) = \min_k \left( \frac{N_k^{\min}}{N_k^{\max}} \right) \in [0, 1], \tag{22}$$

where $N_k^{min}$ and $N_k^{max}$ represent the number of instances in the smallest and the largest (in size) protected groups in cluster $\pi_k$, respectively. MNCE is defined as:

$$\text{MNCE} = \frac{\min_k \left( -\sum_i \frac{|\mathcal{G}_i \cap \pi_k|}{|\pi_k|} \log \frac{|\mathcal{G}_i \cap \pi_k|}{|\pi_k|} \right)}{-\sum_i \frac{|\mathcal{G}_i|}{n} \log \frac{|\mathcal{G}_i|}{n}} \in [0, 1]. \tag{23}$$

For all metrics, the larger the value is, the better the result is.

## 4.3 Experimental Results

Table 2 shows the results of multi-view clustering methods, where the red and blue texts indicate the best and second-best results, respectively. As introduced before, Fair-MVC can only handle data sets with two views, and thus it can only run results on Credit Card and Hcv data sets. It shows that our method outperforms other multi-view clustering methods w.r.t. the fairness metrics (i.e., Bal and MNCE) on all data sets. Even compared with the fair multi-view clustering method Fair-MVC, ours still achieves fairer results. It well shows the superiority of our fairness-aware regularized term, demonstrating our motivation for fairness. Besides this, our method is still comparable with or even better than other methods w.r.t. clustering performance (i.e., ACC and NMI) on most data sets.

Table 3 shows the comparison results with the single-view fair clustering methods. Considering the high spatial complexity in the initial stage of FFC, we randomly select some permutations from all initialized permutation combinations. It can be seen that our method performs better than the single-view fair clustering

**Table 2: Comparison results with multi-view clustering methods on all data sets. Red texts indicate the best results, and blue texts indicate the second-best results.**

| Data sets | | AMGL | AWP | CGD | COMVSC | SMSC | OPLFMVC | CGL | EMVGC | RCAGL | Fair-MVC | FMSC |
|---|---|---|---|---|---|---|---|---|---|---|---|---|
| Yale | ACC | 0.394 | 0.606 | 0.533 | 0.702 | 0.709 | 0.721 | 0.697 | 0.448 | 0.721 | - | 0.761 |
| | NMI | 0.475 | 0.641 | 0.577 | 0.755 | 0.718 | 0.740 | 0.759 | 0.517 | 0.787 | - | 0.840 |
| | Bal | 0 | 0 | 0 | 0 | 0 | 0 | 0 | 0 | 0 | - | 0.100 |
| | MNCE | 0 | 0 | 0 | 0 | 0 | 0 | 0 | 0 | 0 | - | 0.580 |
| COIL | ACC | 0.794 | 0.462 | 0.793 | 0.657 | 0.826 | 0.470 | 0.556 | 0.398 | 0.657 | - | 0.804 |
| | NMI | 0.883 | 0.652 | 0.879 | 0.755 | 0.918 | 0.640 | 0.731 | 0.552 | 0.808 | - | 0.890 |
| | Bal | 0.531 | 0.700 | 0.531 | 0.516 | 0.531 | 0.636 | 0.583 | 0.428 | 0.428 | - | 0.764 |
| | MNCE | 0.931 | 0.977 | 0.931 | 0.925 | 0.931 | 0.964 | 0.949 | 0.881 | 0.881 | - | 0.987 |
| Jaffe | ACC | 0.826 | 0.787 | 0.860 | 0.861 | 0.849 | 0.751 | 0.770 | 0.511 | 0.530 | - | 0.865 |
| | NMI | 0.826 | 0.78 | 0.832 | 0.851 | 0.815 | 0.781 | 0.838 | 0.554 | 0.633 | - | 0.842 |
| | Bal | 0 | 0 | 0 | 0 | 0 | 0 | 0 | 0 | 0 | - | 0.500 |
| | MNCE | 0.356 | 0.562 | 0.842 | 0.872 | 0.556 | 0.645 | 0.564 | 0.516 | 0.354 | - | 0.987 |
| ORL | ACC | 0.627 | 0.782 | 0.537 | 0.527 | 0.74 | 0.737 | 0.845 | 0.487 | 0.655 | - | 0.840 |
| | NMI | 0.780 | 0.875 | 0.699 | 0.712 | 0.884 | 0.863 | 0.921 | 0.597 | 0.839 | - | 0.912 |
| | Bal | 0 | 0.111 | 0 | 0 | 0.147 | 0.140 | 0.080 | 0 | 0.114 | - | 0.200 |
| | MNCE | 0 | 0.369 | 0 | 0 | 0.390 | 0.382 | 0.361 | 0 | 0.372 | - | 0.500 |
| Scene | ACC | 0.327 | 0.297 | 0.422 | 0.192 | 0.401 | 0.315 | 0.419 | 0.370 | 0.334 | - | 0.390 |
| | NMI | 0.302 | 0.260 | 0.380 | 0.133 | 0.404 | 0.288 | 0.383 | 0.335 | 0.294 | - | 0.401 |
| | Bal | 0 | 0.809 | 0.788 | 0 | 0.771 | 0.822 | 0.717 | 0.806 | 0 | - | 0.881 |
| | MNCE | 0 | 0.992 | 0.990 | 0 | 0.988 | 0.993 | 0.980 | 0.991 | 0 | - | 0.997 |
| Har | ACC | 0.366 | 0.395 | 0.565 | 0.353 | 0.557 | 0.511 | 0.417 | 0.552 | 0.550 | - | 0.628 |
| | NMI | 0.399 | 0.353 | 0.603 | 0.368 | 0.605 | 0.380 | 0.398 | 0.525 | 0.565 | - | 0.609 |
| | Bal | 0 | 0 | 0 | 0 | 0 | 0 | 0 | 0 | 0 | - | 0.180 |
| | MNCE | 0.261 | 0.916 | 0.221 | 0.325 | 0.288 | 0.893 | 0.277 | 0.865 | 0.478 | - | 0.971 |
| Credit Card | ACC | 0.380 | 0.371 | 0.371 | 0.286 | 0.261 | 0.297 | 0.252 | 0.366 | 0.266 | 0.396 | 0.382 |
| | NMI | 0.151 | 0.122 | 0.134 | 0.064 | 0.043 | 0.050 | 0.042 | 0.155 | 0.058 | 0.246 | 0.160 |
| | Bal | 0.547 | 0.626 | 0.553 | 0.527 | 0.113 | 0.545 | 0.234 | 0.576 | 0.524 | 0.591 | 0.687 |
| | MNCE | 0.96 | 0.985 | 0.962 | 0.952 | 0.613 | 0.959 | 0.644 | 0.970 | 0.951 | 0.972 | 0.999 |
| Hcv | ACC | 0.375 | 0.361 | 0.253 | 0.378 | 0.413 | 0.400 | 0.374 | 0.317 | 0.457 | 0.462 | 0.487 |
| | NMI | 0.019 | 0.029 | 0.021 | 0.093 | 0.115 | 0.106 | 0.028 | 0.025 | 0.120 | 0.129 | 0.133 |
| | Bal | 0 | 0 | 0 | 0.126 | 0.286 | 0.012 | 0 | 0 | 0.312 | 0.596 | 0.625 |
| | MNCE | 0 | 0 | 0 | 0.772 | 0.843 | 0.413 | 0 | 0 | 0.953 | 0.994 | 0.998 |

**Table 3: Comparison with fair clustering methods. Red texts indicate the best results**

| Methods | Yale | | | | COIL | | | |
|---|---|---|---|---|---|---|---|---|
| | ACC | NMI | Bal | MNCE | ACC | NMI | Bal | MNCE |
| SpFC | 0.714 | 0.771 | 0.095 | 0.561 | 0.840 | 0.935 | 0.526 | 0.924 |
| VFC | 0.690 | 0.730 | 0.100 | 0.580 | 0.729 | 0.799 | 0.636 | 0.964 |
| FFC | 0.739 | 0.789 | 0.083 | 0.516 | - | - | - | - |
| FMSC | 0.761 | 0.840 | 0.100 | 0.580 | 0.804 | 0.890 | 0.764 | 0.987 |

| Methods | Jaffe | | | | ORL | | | |
|---|---|---|---|---|---|---|---|---|
| | ACC | NMI | Bal | MNCE | ACC | NMI | Bal | MNCE |
| SpFC | 0.861 | 0.824 | 0.111 | 0.675 | 0.641 | 0.784 | 0 | 0 |
| VFC | 0.751 | 0.797 | 0.333 | 0.927 | 0.607 | 0.777 | 0 | 0 |
| FFC | 0.798 | 0.865 | 0.200 | 0.945 | - | - | - | - |
| FMSC | 0.865 | 0.842 | 0.500 | 0.987 | 0.840 | 0.912 | 0.200 | 0.500 |

| Methods | Scene | | | | Har | | | |
|---|---|---|---|---|---|---|---|---|
| | ACC | NMI | Bal | MNCE | ACC | NMI | Bal | MNCE |
| SpFC | 0.308 | 0.296 | 0.647 | 0.966 | 0.547 | 0.587 | 0 | 0 |
| VFC | 0.274 | 0.284 | 0.681 | 0.973 | 0.578 | 0.583 | 0 | 0.672 |
| FFC | 0.317 | 0.329 | 0.250 | 0.721 | 0.670 | 0.564 | 0.079 | 0.966 |
| FMSC | 0.390 | 0.401 | 0.881 | 0.997 | 0.628 | 0.609 | 0.180 | 0.971 |

| Methods | Credit Card | | | | Hcv | | | |
|---|---|---|---|---|---|---|---|---|
| | ACC | NMI | Bal | MNCE | ACC | NMI | Bal | MNCE |
| SpFC | 0.299 | 0.103 | 0.602 | 0.978 | 0.474 | 0.102 | 0.501 | 0.981 |
| VFC | 0.378 | 0.235 | 0.576 | 0.970 | 0.396 | 0.039 | 0 | 0 |
| FFC | 0.388 | 206 | 0.566 | 0.967 | 0.341 | 0.044 | 0.084 | 0.409 |
| FMSC | 0.382 | 0.160 | 0.687 | 0.999 | 0.487 | 0.133 | 0.625 | 0.998 |

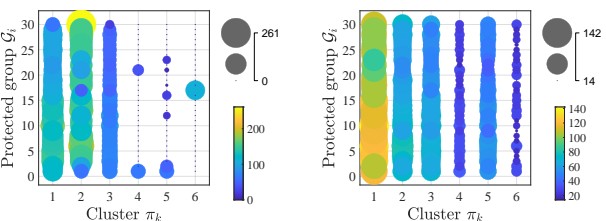

(a) Without the fairness-aware regularized term. (b) With the fairness-aware regularized term.

**Figure 2: The distribution of the protected groups in each cluster on Har data set. The size and color of a bubble denote the number of instances.**

methods w.r.t. both clustering performance and fairness on most data sets. It is because we effectively ensemble the information from multiple views to improve the final clustering performance, which is consistent with the motivation of multi-view learning. Even when compared with the fairness metrics, ours still outperforms these fair methods, which shows the superiority of our designed fairness-aware regularized term.

To further show the effects of our fair-aware regularized term, we visualize the distribution of instances in each protected group and

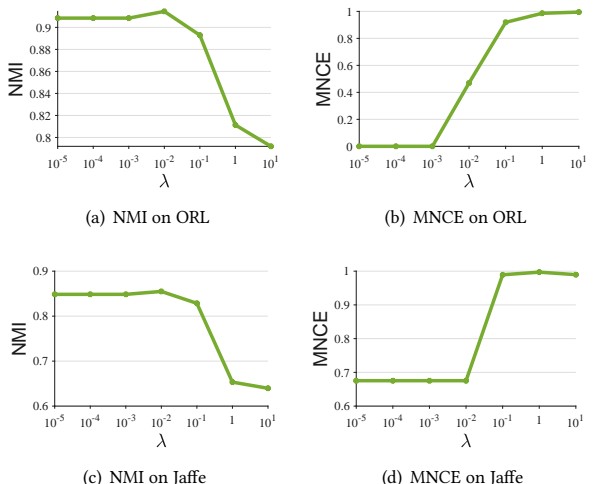

(a) NMI on ORL  (b) MNCE on ORL

(c) NMI on Jaffe  (d) MNCE on Jaffe

**Figure 3: NMI and MNCE on ORL and Jaffe data sets w.r.t. different values of $\lambda$.**

each cluster. Figure 2 shows the visualization results on the Har data set. The X-axis and Y-axis denote the clusters and protected groups, respectively. The size and color of a bubble denotes the number of instances in a protected group and a cluster $|\pi_k \cap \mathcal{G}_i|$. Figure 2 (a) and (b) visualize the results without and with the fairness-aware regularized term, respectively. We can find that the distribution of protected groups in clusters 4,5, and 6 in Figure 2 (a) is very imbalanced, which means the results without the fairness-aware regularized term are unfair according to Definition 1. In Figure 2(b), we can see that the distribution of protected groups in all clusters is balanced, which means the results are much fairer than the one without the regularized term. This result well demonstrates the effectiveness of our proposed fairness-aware regularized term.

## 4.4 Ablation Study

We conduct the ablation study to show the effectiveness of our fairness-aware regularized term and the one-stage clustering strategy. Specifically, we denote **FMSC-nf** as FMSC without the fairness-aware regularized term, which means we set $\lambda = 0$; we denote **FMSC-ts** as the two-stage method that obtains a consensus spectral embedding first, and then discretizes the embedding with spectral rotation together with our fairness-aware regularized term to obtain final discrete clustering results. Table 4 shows the results. It shows that, compared with the two-stage method, our FMSC achieves better clustering performance w.r.t. ACC and NMI. This is because spectral clustering is only used in the first stage of FSMC-ts, and in the second stage of FSMC-ts, it does not control the clustering accuracy. The fairness of FSMC-ts is comparable with our FSMC because in the second stage of FSMC-ts, it also directly uses our fairness-aware regularized term to control the fairness. Despite this, our FSMC also outperforms it on some data sets, demonstrating the superiority of our one-stage clustering strategy.

When compared with FSMC-nf, FSMC performs better w.r.t. the fairness (i.e., Bal and MNCE) on all data sets. It well demonstrates the effectiveness of our designed fairness-aware regularized term. However, it is interesting to see that, on some data sets, such as Yale

**Table 4: Ablation Study on all data sets. Red texts indicate the best results**

| Methods | Yale | | | | COIL | | | |
|---|---|---|---|---|---|---|---|---|
| | ACC | NMI | Bal | MNCE | ACC | NMI | Bal | MNCE |
| FMSC-nf | 0.773 | 0.851 | 0 | 0 | 0.809 | 0.913 | 0.531 | 0.936 |
| FMSC-ts | 0.732 | 0.806 | 0.100 | 0.580 | 0.775 | 0.831 | 0.821 | 0.995 |
| FMSC | 0.761 | 0.840 | 0.100 | 0.580 | 0.804 | 0.890 | 0.764 | 0.987 |
| Methods | Jaffe | | | | ORL | | | |
| | ACC | NMI | Bal | MNCE | ACC | NMI | Bal | MNCE |
| FMSC-nf | 0.850 | 0.831 | 0 | 0.670 | 0.830 | 0.905 | 0.102 | 0.469 |
| FMSC-ts | 0.812 | 0.783 | 0.449 | 0.971 | 0.803 | 0.872 | 0.192 | 0.631 |
| FMSC | 0.865 | 0.842 | 0.500 | 0.987 | 0.840 | 0.912 | 0.200 | 0.500 |
| Methods | Scene | | | | Har | | | |
| | ACC | NMI | Bal | MNCE | ACC | NMI | Bal | MNCE |
| FMSC-nf | 0.398 | 0.412 | 0.840 | 0.994 | 0.637 | 0.680 | 0 | 0 |
| FMSC-ts | 0.336 | 0.348 | 0.912 | 0.997 | 0.538 | 0.527 | 0.116 | 0.923 |
| FMSC | 0.390 | 0.401 | 0.881 | 0.997 | 0.628 | 0.609 | 0.180 | 0.971 |
| Methods | Credit Card | | | | Hcv | | | |
| | ACC | NMI | Bal | MNCE | ACC | NMI | Bal | MNCE |
| FMSC-nf | 0.390 | 0.165 | 0.550 | 0.964 | 0.258 | 0.020 | 0 | 0 |
| FMSC-ts | 0.334 | 0.125 | 0.681 | 0.998 | 0.445 | 0.092 | 0.625 | 0.998 |
| FMSC | 0.382 | 0.160 | 0.687 | 0.999 | 0.487 | 0.133 | 0.625 | 0.998 |

and COIL, when removing the fairness regularized term, FMSC-nf achieves better performance w.r.t. ACC and NMI compared to our FMSC. It is reasonable. Notice that, when computing ACC and NMI, we need to use the ground truth of the data. However, on some data sets, the ground truth may be naturally unfair. When we impose the fairness-aware regularized term on these data sets, we can only achieve a trade-off between accuracy and fairness, which means we should sacrifice the clustering accuracy to achieve fairness.

## 4.5 Efficiency Results

Due to the limited space, we show the convergence curves of our method and the running time of all methods on all data sets in the Appendix. Our method typically converges within 10 iterations. In terms of running time efficiency, our method is comparable with state-of-the-art methods, and even faster than many methods such as COMVSC and SMSC.

## 4.6 Hyper-parameter Study

Figure 3 shows the NMI and MNCE on ORL and Jaffe data sets with different values of $\lambda$ within $\left[10^{-5}, 10^{1}\right]$. The results on other data sets are similar. As $\lambda$ increases, fairness (i.e., MNCE) gradually improves, while clustering performance (i.e., NMI) may decrease, which is consistent with our previous analysis. $\lambda$ controls the trade-off between the clustering accuracy and fairness and when $\lambda = 0.1$ we can obtain a relatively good trade-off.

## 5 CONCLUSION

This paper proposed a novel one-stage fair multi-view spectral clustering method. We offered a new observation and explanation of fairness from the graph theory viewpoint to construct a relation between fairness and the average degree. Based on this observation, we designed a new fairness-aware regularized term that has the same form as spectral clustering. Therefore, we can naturally and seamlessly plug it into multi-view spectral clustering, leading to our FMSC method. Extensive experiments have been conducted to demonstrate the superiority of our proposed method. The ablation study also demonstrates the effectiveness of our designed fairness-aware regularized term.

## ACKNOWLEDGMENTS

This work is supported by the National Natural Science Foundation of China grants 62176001 and 62376146, and Natural Science Project of Anhui Provincial Education Department grants 2023AH030004.

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
