# OpenReview forum: "One-Stage Fair Multi-View Spectral Clustering"
_acmmm.org/ACMMM/2024/Conference — MM2024 Poster_

### Official Review · Reviewer_a7A6 · 2024-05-01

**Rating:** 6
**Confidence:** 4

**Summary:**

Multi-view clustering plays a critical role in multimedia and machine learning, with multi-view spectral clustering standing out as a highly effective approach. Yet, a significant oversight in existing methodologies is their neglect of fairness in clustering outcomes, potentially leading to discriminatory results. Addressing this gap, the paper introduces an innovative Fair Multi-view Spectral Clustering (FMSC) strategy that integrates fairness into the clustering process. This approach is underpinned by a novel concept of fairness rooted in graph theory, specifically linking fairness to the average degree within a graph. By crafting a fairness-aware regularization term analogous to the spectral clustering’s ratio cut, the method embeds fairness considerations directly into the multi-view spectral clustering framework. As a result, the FMSC algorithm facilitates the attainment of fair clustering outcomes in a straightforward, single-stage process that eliminates the need for any post-processing. The superiority of FMSC in achieving greater fairness is substantiated through detailed comparative experiments against leading fair and multi-view clustering techniques, highlighting its effectiveness.

**Strengths:**

Innovative Approach to Fairness: FMSC introduces a novel perspective on embedding fairness within multi-view spectral clustering by drawing on concepts from graph theory. The innovative link between fairness and the graph’s average degree represents a forward-thinking approach to equitable machine learning.

Integration of Fairness into Clustering Process: The development of a fairness-aware regularization term, analogous to the ratio cut, allows FMSC to seamlessly incorporate fairness directly into the clustering algorithm. This ensures that fairness is not an afterthought but a fundamental component of the clustering process.

Efficiency and Simplicity: By embedding fairness considerations into the clustering algorithm itself, FMSC eliminates the need for post-processing to achieve fair outcomes. This one-stage process simplifies the methodology while ensuring that the final clustering results are inherently fair.

Empirical Validation: Extensive experiments demonstrating the superiority of FMSC over existing state-of-the-art methods in achieving fairness provide empirical support for its effectiveness. These results affirm the method’s capability to enhance fairness in multi-view clustering outcomes.

**Limitations:**

Complexity of Fairness Perspective: While FMSC’s new perspective on fairness is innovative, it might also introduce complexity in terms of understanding and implementing the fairness-aware regularized term, especially for practitioners who are not deeply familiar with graph theory.

Potential for Over-Regularization: The introduction of a fairness-aware regularization term may lead to over-regularization in certain scenarios, potentially impacting the clustering performance regarding other critical metrics like accuracy or cohesion.

Possible Computation Overheads: Integrating additional fairness considerations into the spectral clustering process could introduce computational overheads, especially on large-scale datasets, potentially making the algorithm less efficient compared to other non-fairness-aware methods.

**Suitability:**

3

---

### Official Review · Reviewer_asMD · 2024-05-22

**Rating:** 3
**Confidence:** 3

**Summary:**

This paper proposes a one-stage Fair Multi-View Spectral Clustering (FMSC) method that integrates a fairness-aware regularized term into multi-view spectral clustering from a graph theory perspective. The approach directly addresses the issue of fairness in clustering results, achieving both high clustering accuracy and fairness without post-processing. The experimental results on various benchmark datasets demonstrate the superiority of the proposed method, especially in terms of fairness.

**Strengths:**

1. The paper is well-structured, and the arguments are well-supported, making it a valuable contribution to the field.

2. The FMSC method introduces the concept of fairness in multi-view clustering from a graph theory perspective, constructing a relationship between fairness and average degree. This new perspective is both theoretically significant and practically valuable.

3. Unlike traditional two-stage methods, the FMSC method directly optimizes the discrete cluster indicator matrix, eliminating the need for post-processing steps. This integration enhances both clustering accuracy and fairness.

**Limitations:**

1. The literature review does not clearly highlight what methods were used to solve what problems; it merely provides a simple introduction to the topic.

2. In line 237, it seems that the proposed method's goal is merely to advance the method in [58], while this paper introduces a new method that is not typical in the community so far. This makes the contribution of this paper less convincing.

3. In line 244, it is not clear why a soft method mitigates fairness.

4. In line 355, regarding the equation, the numerator and denominator are not exactly the same, and they do not seem to equate to 1.

5. Concerning the proof, based on my understanding, it appears to prove that only when the equation in eq.10 holds can it convert the fairness definition into the degree aspect. This does not lead to the claims in line 375. If not, the explanation of eq.10 could be clearer and more detailed.

6. When choosing the protected data, what are the criteria? Are they based on label information or specific image characteristics?

7. With the proof, it is based on the complete sub-graph, while in the k-nn graph, it seems not easy to obtain such a complete graph structure. Does this affect the application of the theoretical proof to the method?

**Suitability:**

2

---

### Official Review · Reviewer_7Siu · 2024-05-23

**Rating:** 6
**Confidence:** 4

**Summary:**

This paper proposes a fair multi-view spectral clustering method. It first designs a fair-aware regularized term from the perspective of graph theory, which constructs a relation between the fairness and average degree. Then it plugs the regularized term into the multi-view spectral framework, leading to the fair multi-view spectral clustering. The authors also conduct extensive experiments to evaluate the clustering performance and fairness. The results show that the proposed method is competitive.

**Strengths:**

-	Fairness is important in practice, and it is quite underexplored in multi-view clustering. It is worthy paying more attention to the fairness.

-	The fairness-aware regularized term is interesting and novel. It is well motivated and constructs a connection between the fairness and the average degree, which is impressive.


-	The experiments are sufficient and convincing. The comparison results and ablation study demonstrate the effectiveness of the fairness-aware regularized term.

-	The paper is technically sound and overall well-written.

**Limitations:**

-	In ablation study, FMSC-ts sometimes performs better than FMSC w.r.t. Fair and MNCE. Why? Does it mean the two-stage method is better than the one-stage method? It needs more detailed explanation.

-	The font size in Figure 4 is a little too small.

-	$\lambda$ is a trade-off between the clustering accuracy and fairness. In practice, how to set $\lambda$ to obtain a good trade-off on a new unlabeled data? I think it is an interesting and practical question and needs more detailed discussion.

**Suitability:**

3

---

### Official Review · Reviewer_52tu · 2024-05-23

**Rating:** 5
**Confidence:** 4

**Summary:**

In this paper, the authors focus on the fairness issue in multi-view spectral clustering. The authors follow a classical definition of fairness and constructs a protected group graph to characterize the fairness. They observe and prove that minimizing the total average degree of the protected group graph will result in the optimal of the fairness. Based on this theoretical analysis, the authors design a fairness-aware regularized term, which has the similar form to the ratio cut, and thus can be easily integrated into the multi-view spectral clustering framework. The extensive experiments show that this term can indeed lead to a fair clustering result.

**Strengths:**

Here are some strengths:
1)	Theorem 1 is interesting and provides a new perspective of fairness from the graph viewpoint. It transfers the fairness problem to the minimizing of average degree on the protected group graph. This idea can be used in many graph based methods.
2)	The design of the fairness-aware regularized term is elegant. It computes the average degree but has the similar form to the ratio cut. Therefore, it can be seamlessly and cleverly integrated into the spectral clustering framework.
3)	The experimental results are good. The proposed method can achieve fairer clustering result but comparable or even better clustering accuracy. It seems that the method can make a good trade-off between the accuracy and fairness.

**Limitations:**

Here are some questions and concerns:
1)	The standard spectral clustering can be optimized very effectively. We just need the eigenvalue decomposition on the Laplacian matrix to obtain the spectral embedding and run kmeans on the spectral embedding. In the proposed method, due to the fairness-aware regularized term, it needs to optimize a discrete combinatorial optimization problem. As we know, the discrete combinatorial optimization is much more difficult. So, what is the advantage of the proposed method compared to the continual optimization such as standard spectral clustering?
2)	Since the paper focuses on the multi-view spectral clustering, in Section 2.1, related work about the multi-view spectral clustering should be introduced in more detail.
3)	In the multi-view fusion, since all $\alpha_i$ is non-negative, why use $\alpha_i^2$ as the weight in the loss function instead of using $\alpha_i$ as the weight directly?

**Suitability:**

3

---

### Meta-Review · Area_Chair_Ef1b · 2024-07-01

**Recommendation:** Accept (Poster)
**Confidence:** 5

**Metareview:**

In this paper, the authors focus on the fairness issue of multi-view spectral clustering. In the paper, the authors proposed a new perspective to address this issue, which can be extended to many graph based methods. After rebuttal and discussion, the paper received three positive recommendations and one borderline negative recommendation (A, A, WA, BR) . Three reviewers strongly vote to accept the paper. According to the novelty and contributions of the paper, I agree with the recommendations of most reviewers, and thus an acceptance decision is made.